# A Quantitative Survey of Bacterial Persistence in the Presence of Antibiotics: Towards Antipersister Antimicrobial Discovery

**DOI:** 10.3390/antibiotics9080508

**Published:** 2020-08-13

**Authors:** Jesus Enrique Salcedo-Sora, Douglas B. Kell

**Affiliations:** 1Department of Biochemistry and Systems Biology, Institute of Systems, Molecular and Integrative Biology, Biosciences Building, University of Liverpool, Crown Street, Liverpool L69 7ZB, UK; dbk@liv.ac.uk; 2Novo Nordisk Foundation Centre for Biosustainability, Technical University of Denmark, Building 220, Kemitorvet, 2800 Kgs. Lyngby, Denmark

**Keywords:** bacteria, antibiotics, resistance, persisters, dormancy, time–kill, membrane transport, membrane transporters, lipopeptides

## Abstract

**Background**: Bacterial persistence to antibiotics relates to the phenotypic ability to survive lethal concentrations of otherwise bactericidal antibiotics. The quantitative nature of the time–kill assay, which is the sector’s standard for the study of antibiotic bacterial persistence, is an invaluable asset for global, unbiased, and cross-species analyses. **Methods**: We compiled the results of antibiotic persistence from antibiotic-sensitive bacteria during planktonic growth. The data were extracted from a sample of 187 publications over the last 50 years. The antibiotics used in this compilation were also compared in terms of structural similarity to fluorescent molecules known to accumulate in *Escherichia coli*. **Results**: We reviewed in detail data from 54 antibiotics and 36 bacterial species. Persistence varies widely as a function of the type of antibiotic (membrane-active antibiotics admit the fewest), the nature of the growth phase and medium (persistence is less common in exponential phase and rich media), and the Gram staining of the target organism (persistence is more common in Gram positives). Some antibiotics bear strong structural similarity to fluorophores known to be taken up by *E. coli*, potentially allowing competitive assays. Some antibiotics also, paradoxically, seem to allow more persisters at higher antibiotic concentrations. **Conclusions**: We consolidated an actionable knowledge base to support a rational development of antipersister antimicrobials. Persistence is seen as a step on the pathway to antimicrobial resistance, and we found no organisms that failed to exhibit it. Novel antibiotics need to have antipersister activity. Discovery strategies should include persister-specific approaches that could find antibiotics that preferably target the membrane structure and permeability of slow-growing cells.

## 1. Introduction

Hobby [1] and Bigger [2] in the 1940s noted the survival of small fractions of bacteria in axenic planktonic cultures of both *Staphylococcus* and *Streptococcus* spp. after sustained exposure to normally bactericidal concentrations of the recently deployed antibiotic penicillin. They were also able to show that this effective ‘differentiation’ of the cultures into sensitive and resistant cells was purely phenotypic in nature; reinoculation of the survivors showed that they were just as susceptible as they had been in the initial cultures. Bigger [2] referred to this surviving fraction of cells as ‘persisters’. The concept of persistence to antibiotics was further elaborated by Greenwood, who detected exponential growth of *E. coli* and *Proteus mirabilis* post exposure to unrelated antibiotics [3] and inferred the involvement of membrane permeability by changing the osmolality of the media [4,5,6]. However, the subsequent interest in such phenotypic tolerance to antibiotics by phenotypic variability was sporadic at best, and was only rekindled many decades later (e.g., [7,8]). The renewed scrutiny has now crucially shown that bacterial persistence to antibiotics can lead directly to antibiotic resistance [9,10,11,12]. This is of great importance since bacterial persisters have been found to cause recalcitrant clinical infections [13,14,15,16,17,18]. Nonetheless, early work implicating membrane transport in bacterial persistence to antibiotics [19] has not attracted the same level of attention.

Around the same time, a related finding of phenotypic bacterial differentiation in laboratory cultures was that the killing of *E. coli* exposed to UV radiation “…is an exponential function of the dose. As observed by Witkin [20] a change in the slope of the curve appears for survivals less than 10^−2^, as if 1 per cent of the bacteria had higher resistance” [21]. Given that penicillin kills only growing cells [22,23], an obvious inference was that the surviving fraction might have escaped death by entering some kind of a dormant or non-growing state, something that is in fact the norm in natural microbial communities [24,25,26,27].

The phenotypes reflecting persistence to antibiotics are another manifestation of the mosaic composition of any given bacterial population, which allows some cells to survive changes in otherwise cytotoxic environments [28]. This individual diversity manifests, for instance, as differences in cell size, macromolecular content, and gene expression in otherwise genetically identical populations [8,29,30,31,32,33]. From an evolutionary point of view, such strategies of phenotypic differentiation may be seen as ‘bet hedging’ [34,35]. Cells better suited to surviving transient exposure to high concentrations of antibiotics are slow-growing cells, which are a minority when the environmental conditions are conducive to rapid cell multiplication (e.g., exponential phase in batch cultures) [7,8,36]. When the surrounding milieu has been exhausted of nutrients, cell growth stagnates, and the cell fraction that persists to antibiotics is somewhat higher (e.g., in the stationary phase in batch cultures) [37,38,39,40]. Any microorganism will find itself in cycles continuously oscillating between these two extremes of cell growth [41,42] and henceforth surviving exposure to antibiotics (or any other toxins) with different degrees of success.

The mechanistic phenomena causing individual cells to be slow growers in an otherwise fast-growing population would be expected to involve systems than lend themselves to stochastic molecular cycles, and necessarily involve at least two partners that serve to ‘flip’ the cells between such growing and non-growing states. A subset of such pairs is known as toxin-antitoxin pairs [43,44,45,46,47,48]. The first of those described was the protein kinase *HipA*, a type II toxin of *E. coli* that inhibits protein translation when stochastically active [49,50]. However, the role of 10 type II toxin-antitoxin systems in antibiotic persisters has been derogated to be the confounded effect of a widespread bacteriophage contamination of lab strains of *E. coli* [51,52] or has had their function reannotated based on new experimental evidence (e.g., *pasT*) [53]. The porin *hokB* can stochastically dimerize in Gram-negative bacteria, which leads to ATP leakage and a metabolic slowdown that increases the tolerance to antibiotics [54,55]. Indeed, lowered levels of ATP are known to prime cells to tolerate stress, such as an exposure to high concentrations of antibiotics [56,57,58]. Another mechanism that fits the cyclic and stochastic nature of phenotypic variability is the build-up and energy-consuming repurposing cycle of protein aggregation through the growth cycle of *E. coli*. This has been shown to exhibit differential kinetics among individual cells. The rates at which cells re-process these protein conglomerates are related to the length of their lag phases and their capacity to survive antibiotics [59,60,61,62,63,64,65]. The growing list of cellular systems with genes that express stochastically now includes enzymes of central carbon metabolism, such as lactate dehydrogenase, and enzymes of the tricarboxylic acid cycle [66,67].

It is clear that the quantitative nature of the data on bacterial persistence to antibiotics should be an asset to exploit. An extensive compilation of the data from the now ubiquitous biphasic killing assay [68], for example, might bring a level of numerical consensus to a field prone to the inconsistencies often observed in work in bacterial physiology and between results from different laboratories. The data compiled here, from a large sample of the numerical information available in the literature, reflect that aim, and draw attention to quantifiable differences in decisive drivers of tolerance to antibiotics, such as the growth phase, antibiotic class, bacterial species, and culture media.

Since these variables can each reflect the physiology of a microbial population, we connected them with the available information on the involvement of membrane permeability in tolerance to antibiotics. We then integrated this with evidence that argues for a causal effect of membrane transport of xenobiotics (e.g., antibiotics), as much as natural nutrient substrates, in persistence to antibiotics. This survey brings into context the fact that as in any other living cell, antibiotics must first interact with and, in most cases, cross the cell membrane(s) in order to reach their target(s) [69,70,71,72,73].

## 2. Results

### 2.1. Data Summary and Praxis

The data extracted covered information for 36 bacterial species (Table 1) and 54 antibiotics (Table 2). The following variables were tabulated together with the percentage (against the control at time zero) of cells growing post-exposure to antibiotics: Species, growth phase, antibiotic, MIC (minimal inhibitory concentration), antibiotic dosage, and time (hours) of exposure to the antibiotic (Appendix A). As expected, *E. coli* had the most numerous records from studies with 32 different antibiotics, followed by *S. aureus* and *Pseudomonas aeruginosa*, with 18 and 16 antibiotics, respectively (Figure 1). The median of the percentage of bacterial cells surviving exposure to nominally lethal concentrations of antibiotics spanned five orders of magnitude from 7 × 10^−4^% in *P. putida* to 100% in *E. faecium* (Figure 2). If we consider species with 20 or more data points in total, this range was narrower, with the lowest level of persistence starting at 0.01% in *A. baumannii*. The multidrug-resistant *S. aureus* MRSA was represented in the top 10 species with an incidence of 5% of cells persistent to antibiotics.

Persister assays are carried out using antibiotics at concentrations above their inhibitory concentrations (commonly reported as MIC, minimal inhibitory concentration). We compared the concentration of antibiotics used against their MICs, as measured in nearly half of the publications: Antibiotics were used at a median value of 40 times their MICs. The range of persisters clustered by antibiotic also spread across five orders of magnitude, starting with colistin at 0.001% and reaching 63% and 71% for erythromycin and metronidazole, respectively (Figure 3). We note that the latter is represented by only two data points.

The frequencies of persisters are known to differ according to the antibiotic used. This plausibly reflects the nature of any active and varied metabolism in these cells [89]. After arranging these antibiotics by class (Figure 4) and by mechanism of action (Figure 5), it is noticeable that the lowest numbers of persisters were observed for those molecules that target the cell membrane while the opposite was the case for antimetabolites, such as antifolates, and for inhibitors of protein biosynthesis, such as macrolides. The former membrane-targeting antibiotics are lipopeptides, such as colistin, daptomycin, and polymyxin B (Table 2). It is noteworthy that the intervals of the data represented in Figure 3, Figure 4 and Figure 5 came from the values from different time points of the biphasic killing assays. This explains in part the distance between the quartiles along the box plots, such as that observed for teicoplanin, among other antibiotics (Figure 3). Nonetheless, using the median values as a measure of the central tendency permits the comparisons drawn here.

### 2.2. Bactericidal vs. Bacteriostatic Antibiotics

It is usual to contrast bacteriostatic antibiotics from those that are bactericidal [90], where we use the operational ability of the target microbes to replicate at any time in the future as a criterion of whether bacteria are or were ‘alive’ or not [91,92,93]. Specifically, as judged by colony formation, bacteriostatic agents are considered merely to cause growth to cease, while bactericidal antibiotics actually decrease the number of organisms whether they are growing or not. The distinction is mainly based on in vitro measurements, and is much less clear cut in clinical situations [90,94,95]. Indeed, there is evidence that bacteriostatic inhibitors may antagonize the activity of bactericidal molecules both in vivo and in vitro [96,97]. This important effect follows straightforwardly from the fact that antibiotics are normally most potent against growing cells (e.g., [22,98,99,100]).

Clearly, in vitro studies are far easier to perform than are those in vivo, and we recognize that the question of persistence is not easy to disentangle from simple bacteriostasis in vitro; organisms that are exposed to a concentration of antibiotic that causes them to cease growth but not to die can by definition regrow as soon as the bacteriostatic antibiotic is removed, and this is (or can be) a purely phenotypic phenomenon. This is why we usually consider persistence in the context solely of normally bactericidal antibiotics. Table 2 and the references therein set out those considered to fall into the two classes. Macrolides, chloramphenicol, and tetracyclines primarily considered bacteriostatic, had levels of persisters equivalent to 11.9%, 8%, and 1.26%, respectively (Figure 4).

Given that bacteriostatic inhibitors do not in fact kill bacteria in vitro, it is of obvious interest to wonder what is different about those that do. To this end, Collins and co-workers (e.g., [101,102,103,104,105,106,107]) showed that the main distinguishing feature was not in fact their targets (e.g., protein synthesis or the cell wall or DNA topoisomerases) but, in significant measure, the extent to which they could cause the cells to produce damaging free radicals [108,109,110]. Such bactericidal activities could kill cells by inducing a state manifesting features equivalent to those accompanying apoptosis in higher organisms [111,112]. Clearly, any analysis of phenotypic persistence needs to recognize that there are many gene products that might be more or less protective against oxidative stress at a single cell level, and so consequently it is not necessarily surprising, as we show here, that its extent may vary quite dramatically.

### 2.3. Growth Phase and Gram Staining Were Related to Significantly Different Levels of Bacterial Persisters to Antibiotics

As anticipated, bacteria in an ostensible stationary phase had a median value for persisters that was circa 30 times higher than the median for persisters found in samples from exponential cultures (Figure 6a, Table 3). This difference is a known trend (e.g., [38,61,113,114,115,116,117,118,119]), but here we are able to state this quantitatively, together with its statistical significance (*p* < 1.1 × 10^−10^) (Figure 7a). The data collated here for bacteria in the exponential phase also included data from cultures denoted in some publications as “mid-exponential” and “late-exponential”. The distributions of the data from the publications using those terms were not different from the distribution of data from just “exponential” cultures.

The type of culture media in which bacteria are growing was expected to be a modifier of the outcome of antibiotic challenge in persister assays. These media were either complex and nutrient rich, or defined with a more limited nutrient composition. Previous studies have shown that *E. coli* growing in M9 minimal media, for instance, have fewer cells surviving a subsequent challenge with 4-quinolones (i.e., ofloxacin and ciprofloxacin) than when growing in complex media [52,120]. However, the far more extensive data collected here showed no differences between complex and defined media, with median values of 0.2% of persister cells in complex media and 0.16% of persisters cells in defined media (Figure 6b and Figure 7b). On the other hand, the median value of persisters was nine-fold higher in Gram-positive bacteria than in Gram-negative species (Figure 6c and Figure 7c). With data from 20 Gram-negative and 16 Gram-positive bacterial species (Table 1) from this sample of publications, this was possibly an unexpected observation and new to this field. Some of the Gram-positive species of bacteria listed are known to be able to sporulate (Table 1), i.e., *Bacillus* and *Clostridium*. However, the data included here for those bacteria originated from time–kill assays that used samples from planktonic cultures in activate growth, implying germinated populations.

### 2.4. Trends of Time–Kill in Relation to Growth Phase, Media Type, and Gram Staining

The data from time–kill assays in antibiotic persistence have usually fitted (or been fitted to) a biphasic curve. As originally modelled for *E. coli* challenged with quinolones, an initial exponentially descending line observed within the first hour of exposure to antibiotics is followed by an asymptotic decline in surviving bacteria that tends to zero [68]. In the data collated here, the first phase of the time of killing was recorded from 10 min, with an exponential killing that continued for up to 10 h. The rest of the recorded data extended up to hundreds of hours. The longest recorded time–kill point was 504 h for *M. tuberculosis* challenged with ethambutol (Appendix A) [121,122]. The differences in the fitted curves of exponential versus stationary persisters were striking (Figure 8a). The persister minimum (time at which the fewest number of persisters was recorded) for stationary cells was 4 h while it took 10 h to reach that point in cells from exponential cultures. At this point (i.e., 10 h), the difference in the number of persisters was two orders of magnitude, 1% versus 0.01% for stationary and exponential cells, respectively.

The fitted line of the time–kill assays as compiled here for all 36 bacterial species and 54 antibiotics showed that cells growing in the exponential phase, on average, did not seem to converge into the expected asymptotic line of the second phase of a biphasic killing assay (Figure 8a). On the contrary, an opposite trend was observed where the percentage of persister cells increased above the minimum 0.01% after 10 h of exposure for the majority of species (24 of the 36 bacterial species) represented in this survey. This incremental trend of persisters was followed by the initiation of a plateau from approximately 80 h of exposure to antibiotics (Figure 8a).

Cells originated from stationary cultures also had a trimodal time–kill curve (but one different from the exponential growing data), whereby beyond the initial 3 h of exponential decline, the proportion of persister cells stabilized at 1% for up to 10 h. Thereafter, the third phase saw the percentage of persisters decline even further from 1% to approximately 0.01% (Figure 8a). Trimodal growth inhibition curves have been observed in the early work on penicillin in *S. faecalis* and on 4-quinolones in *E. coli*, with the antibiotic concentration as the independent variable [123,124]. It has also been observed in time–kill assays with β-lactams in *E. coli* [5], and in other similar assays where bacterial growth was re-initiated after deactivating the antibiotic using penicillinase [3]. Although these early findings do not seem to be acknowledged in the modern literature, they were recapitulated in the data collected here, which derived from work published since 2003. They plausibly reflect cells periodically emerging from non-growing states to assess conditions, and then being killed off if unfavorable.

If the data are clustered by media type, the comparable levels of persisters between defined and complex media diverged from each other beyond 100 h. Cells originated from cultures in defined media had, at the longest times recorded, a percentage of persisters approximately 20 times higher than those cells that originated from cultures in complex media: 2% and 0.1%, respectively (Figure 8b). When the data were grouped by Gram-staining properties, Gram-positive bacteria had a monotonic trend in the time–kill assay while a triphasic curve was seen again, this time for the fitted data, for the Gram-negative bacteria (Figure 8c). In Gram-negative bacteria, the initial exponential kill phase was followed by an incremental number of cells surviving after 10 h of exposure to antibiotics from approximately 0.01% to 0.9% of persister cells. After 100 h, a further exponential decline reduced the surviving cells to fractional percentages (10^−4^) (Figure 8c).

### 2.5. The Paradoxical Effect of Concentration on Bacterial Killing by Some Antibiotics

The most commonly used antibiotics were ciprofloxacin, ampicillin, ofloxacin, gentamicin, vancomycin, and tobramycin, selected as antibiotics with 100 or more data points of the full data set (Figure 3). These six antibiotics were used for the time–kill assays for 28 different bacterial species. We then wished to explore the effect of incremental concentrations of antibiotics on the number of surviving cells, with particular attention to the paradoxical trend of higher levels of persisters as the concentration of antibiotics increases [125]. Four antibiotics had a sufficient amount of data for the polynomial regression fitting (materials and methods): ampicillin, ciprofloxacin, gentamicin, and ofloxacin.

The challenge of ampicillin showed a decline in persister levels proportional to the concentration of antibiotic from 100 µg/mL and higher (Figure 9a). However, the fluoroquinolones ciprofloxacin (Figure 9b) and ofloxacin (Figure 9d) induced an increase in the number of persister bacteria when used at more than 0.15 and 0.5 µg/mL, respectively. This is of practical importance since the concentrations at which these two antibiotics are frequently used in time–kill assays are higher: 1 µg/mL for ciprofloxacin and 5 µg/mL for ofloxacin (Appendix A and pointed up by the vertical arrows in Figure 9). When used at these high concentrations, ciprofloxacin killed 10-fold fewer bacteria than at the lower concentration of 0.15 µg/mL. In the case of ofloxacin, three-fold fewer bacteria were killed when used at 5 µg/mL than when used at the lower concentration of 0.5 µg/mL. From this group of antibiotics frequently used in time–kill assays, gentamicin appeared to be the only antibiotic to have been frequently used at the concentration that killed the most bacteria and whereby the lowest number of persisters are observed (Figure 9c).

### 2.6. Top Six Bacteria

The top six bacteria ranked by the number of different antibiotics tested for persistence were *E. coli*, *S. aureus*, *P. aeruginosa*, *M. tuberculosis*, *A. baumannii*, and *S. aureus* MRSA (Figure 9 and Figure 10). Importantly, this list includes three of the six species prone to developing rapid antibiotic resistant (ESKAPE) [12,126]: *S. aureus*, *A. baumannii*, and *P. aeruginosa*. The other three ESKAPE species were also represented in this survey although with fewer antibiotics: *E. faecium*, *K. pneumoniae*, and *Enterobacter* spp. (*E. aerogenes*) (Appendix A). The emphasis on these bacteria has been noted and reviewed for *E. coli*, *S. aureus*, *P. aeruginosa*, and *M. tuberculosis* [127]. The data segregated by growth phase mirror the general trend of cells from stationary cultures surviving the challenge to antibiotics better than those that originated from exponentially growing cultures (Figure 9). By visualizing the data from these individual species, it was obvious that studies seemed to concentrate on one or the other growth phase. Thus, for instance, the results for some species, such as *P. aeruginosa*, were primarily from stationary growing cultures while the rest of this group of bacteria had been tested mainly in exponential growth.

A similar and more pronounced bias was found for the influence of the type of culture media on the outcome of antibiotic challenge (Figure 11). The majority of research for these six species of bacteria was carried out with cells growing in complex media. The data from some antibiotics that were studied with both defined as well as complex media showed *E. coli* cells growing in defined media tolerated kanamycin better than cells growing in complex media, with the opposite trend observed for ciprofloxacin (Figure 11). Similar scenarios were shown for rifampicin and isoniazid in *M. tuberculosis*, although the differences in the median values were not significant in either *E. coli* or *M. tuberculosis* (Figure 11). Outside these cases, the data from these species substantiated the general trend, with no difference in the incidence of cells persistent to antibiotics between cells that originated from complex media versus defined media.

### 2.7. Cheminformatics of Antibiotics Represented in This Survey and Fluorophores Transported by E. coli

The data analyzed here provided the opportunity to compare the chemical space of those 54 antibiotics used on bacterial persistence studies with molecules known to be transported by bacteria. Specifically, fluorescent molecules that are found to accumulate in *E. coli* [128] have great potential to study membrane transport, a process of paramount relevance in the study of antimicrobials. We recently reported 47 fluorophores as being able to enter *E. coli* with fluorescent signals two-fold or higher than the background controls [128]. At first glance, important benchmarks provided confidence in the cluster analysis. Antibiotics, such as penicillin and cephalosporins, clustered together as well as did aminoglycosides. Also grouped together were related fluorophores, such as fluorescein derivatives (i.e., 5′-carboxyfluorescein, 2′,5′-dicarboxyfluorescein, fluorescein) as well as the thiazoles neutral red, acridine orange, pyronin Y, methylene blue, and oxazine 1.

The relevance of this structure similarity analysis became apparent with the antibiotic-fluorophore groups observed in Figure 12. Three clusters were found with a Tanimoto similarity [129,130,131] of 0.77 or higher. The sulphonated fluorophores amaranth, sulphorhodamine B, and pyranine clustered with the aminoglycosides streptomycin and amikacin. Another group contained the xanthene dyes eosin Y, calcein, rhodamine 700, and rhodamine 800 included also tetracycline and doxycycline. A third cluster contained trypan blue with rifampicin and daptomycin (Figure 11). Thus, three different classes of antibiotics found a structural cognate in the form of fluorescent molecules known to be transported across the cell membranes of *E. coli* [128].

## 3. Discussion

Bigger and contemporaries [1,2,88] had the exceptional insight of using an ordinary term that embodies with great depth the scientific phenomena that they observed for the first time. A fractional number of bacteria that survive lethal concentrations of antibiotics were named persisters [132]. To persist, these cells would be presumed to have distinctive traits pre-exposure and would emerge as sole colonizers in the next round of growth when the antibiotic dissipates [3]. Those cells, however, retain genotypic sensitivity to the given antibiotic and most of their progeny will be killed if exposure to the antibiotic reoccurs. Nonetheless, and again a connotation given by the term, under iterative exposure, persisters become gradually more tolerant to the stressor, and eventually resistant to it [9,10,11,12,133,134].

The anticipated distinctive traits that allow bacteria to endure the lethality of high concentrations of antibiotics are derived from the variability of cell growth, whereby slow-growing [32,135]—possibly non-dividing [136]—cells of a population arise by the stochastic nature of molecular switches of cell metabolism in bacterial populations in rapid growth (type II persisters) [8,137,138]. A different source of cells that survive the presence of high concentrations of antibiotics are bacteria that find themselves under the eventual shortage of nutrients or have various other sources of stress, also denoted as type I persisters [137,138]. While type II persisters can be regarded as deploying bet-hedging strategies, type I persisters are cells that deploy strongly evolutionarily selected responses against expected stress pressures [32,108,139,140,141,142]. A well-known case among the latter is nutrient scarcity and the associated responses [40,58,139,143], such as what occurs in the case of the stationary growth phase of a bacterial batch culture, which we showed here generates approximately 30-fold more persisters than do the exponentially growing populations. With up to a third of cells in stationary cultures surviving lethal concentrations of antibiotics (Table 3), a significant proportion of cells in wild populations—where limited sources are the norm—are expected to be able to mirror this high level of tolerance to stress. There is in effect an epigenetic memory of individuality in growing cells [59,62].

As a novel observation, Gram-positive bacterial persistence to antibiotics surpassed that of Gram-negative bacteria by an order of magnitude. Since the survival in the face of lethal concentrations of antibiotics is compiled here for different antibiotics and different bacterial species, an intrinsic mechanism, such as the distinctive cell wall of the Gram-positive bacteria [144,145], seems to be at play to enable a more effective protection than in their Gram-negative counterparts. The peptidoglycan wall of Gram-positive bacteria (which may be ~13 times thicker than in Gram-negative bacteria [146]) is functionalized with lipidic complexes of teichoic acid regularly attached to *N*-acetylmuramic acid residues, which makes the total mass of teichoic acid derivatives approximately 60% of the cell wall mass in *B. subtilis* and *S. aureus* [147]. Accordingly, impaired biosynthesis of teichoic acids—which have a negatively charged backbone—renders *S. aureus* MRSA sensitive to β-lactam antibiotics [148]. In this regard, new antibiotics, such as teixobactin, which impair the assembly of the Gram-positive cell wall by binding lipid substrates, have led to great expectations [146] that seem to be well grounded based on the findings of this survey.

We attribute the increased number of persisters observed at time points beyond 80 h in time–kill assays to the reduced stability of antibiotics at 37°C for those lengths of time [149]. This is particularly noticeable when the concentrations of antibiotics used are near the MIC for a given bacterial species and antibiotic pair [122,150,151,152,153]. Levofloxacin had to be present at 4-fold its MIC for *V. vulnificus* to avoid re-growth throughout the time–kill assay [150], ceftibuten had to be at 100-fold the MIC for the same reason in *Enterobacteriaceae* [151], and similarly, for gentamicin and penicillin in *S. faecalis* and *S. aureus*, respectively, with re-growth at 13 h when antibiotics were at their MICs [122,153]. In an in vitro pharmacokinetic model with *E. coli* at 1 h of antibiotic elimination time, growth was observed at the MIC, with a bactericidal effect only when the antibiotic was at 10-fold its MIC [152].

Unexpected growth can also be observed at antibiotic concentrations above their inhibitory concentrations—MIC—in what has been known as paradoxical growth or the Eagle effect [125,154] and seen here in three of the four antibiotics most used in time–kill experiments (Figure 9). The exposure to the two fluoroquinolones ciprofloxacin and ofloxacin, and the aminoglycoside gentamicin produce an increasing number of persisters above 0.15, 0.5, and 0.75 µg/mL, respectively. Examples of an in vitro more-drug-kills-fewer Eagle effect include vancomycin against *C. difficile* [155] and fluoroquinolones against *M. smegmatis* [156]. An in vivo model illustrated this effect in *C. diphtheriae* treated with amoxicillin [157]. Not every bacteria/antibiotic pair assessed has shown this trend and the reports vary, for instance, for β-lactams [158]. In the light of these findings, it is logical to include as a prerequisite the optimal concentration of antibiotic for a persister assay so the survivors are not an overestimation and ultimately not an unprincipled reflection of persistence. This is particularly so since the relation between persistence to antibiotics and the Eagle effect is unknown, although there have been attempts to draw similarities [154].

Antibiotic persistence is a prominent example of the interdependence of deterministic and stochastic mechanisms in the bacterial life cycle [159]. A case in point with accumulating evidence is the aggregation of endogenous proteins and its proportionality to the rate of cell growth (i.e., individual bacteria epigenetic memory [63]) throughout a period of rapid proliferation [160]. Necessary but much less predictable is the asymmetric segregation among the daughter cells of these protein aggregates [62] (which are essentially damaged components). A second layer of stochasticity is introduced by the variability of individual cells in repairing and recycling these aggregates. Bacteria slow at recycling protein aggregates in the lag phase tend to have higher proportions of cells persistent to antibiotics than their more competent siblings [60,61,63]. A similar scenario is the accumulation of glycogen in bacteria during rapid growth, which then fuels cell growth in lag phases with significant interindividual variability [161,162]. However, a potential link of the latter with bacterial persistence to antibiotics has not been explored. The cell-to-cell variability in the cyclic aggregation and reuse of macromolecular cell components along the bacterial life cycle is of particular relevance in the study of persisters. For instance, the recycling of endogenous protein aggregates and the number of persisters throughout the lag phase follow the same long tail exponential decay kinetics [64,163].

The data compiled here indicate that bacteria have fewer persisters to antibiotics that directly interact with and target the cell membrane (i.e., lipopeptides and mitomycins). This is consistent with the body of evidence that cell membranes are a battleground for the asymmetrical partitioning of cytoplasmic contents in cell division, a prevailing source of fluctuation in the bacterial cellular make-up. Diaminopimelic acid is an intermediary of amino acid and peptidoglycan metabolism, which accumulates into the old poles of *E. coli*, as part of the model of asymmetric elongation and division of the cell wall [164]. The proteinaceous content is heterogenous along the periplasmic space of the long axis and poles of *E. coli* [165]. As the medium acidifies during rapid proliferation, rod-shaped bacteria inheriting the older pole divide more slowly than do the cells inheriting the newer pole [166]. An example of specific membrane transporters differentially captured in this old and new pole lineage diversification is the Ser chemoreceptor prompt [162,166,167,168,169]. The segregation of damaged cargo can also be drastically addressed by bacteria producing small spherical non-proliferating (lack of chromosomal DNA) minicells. Bacterial strains vary in their capacity to generate minicells and those that do tend to be more tolerant to antibiotic-induced stress [170].

Membrane transport has been an underlying aspect of bacterial survival to lethal concentrations of antibiotics since the initial reports of this phenomenon [6]. This was complemented by the finding that *Enterobacter* survival to ceftriaxone [171] and *P. aeruginosa* to beta-lactams [172] is mediated by decreased expression of outer membrane proteins (OMPs). Further examples of enhancement of persisters are bacteria with increased activity of effluxer systems, such as TolC [173] and *E. coli* with an impaired BtsSR/YpdaB pyruvate sensing system [174]. Crucially, there is also precedent of modifications in membrane transport systems that sensitize bacteria to the bactericidal effects of antibiotics. Inactivation of phoU (Pho transporter) induces a metabolic hyperactivity that renders *E. coli* and *M. tuberculosis* more sensitive to antibiotics and other stressors [175,176]. Similarly, the induction of transport of exogenous L-tryptophan makes *E. coli* less tolerant to aminoglycosides [177]. Inactivation of ATP-dependent high-affinity potassium (K(+)) transporter in *M. tuberculosis* causes membrane hyperpolarization, increased intracellular ATP, and reduced persisters to rifampicin [178]. The strongest case might be presented by the anthelmintic bithionol, which has been brought forward as a candidate for repurposing as clinically bactericidal, with selective activity against *S. aureus* MRSA [179]. Bithionol is stated to embed selectively into bacterial (and not mammalian) lipid bilayers [179], increasing membrane permeabilization and membrane fluidity. Together with the data from the effects of other membrane-active antimicrobial agents, such as nTZDpa [180], the work collated here presents a strong case for membrane-active compounds to carry the highest antipersister potency.

The development of platforms for antipersister antimicrobials discovery is a vital necessity. This involves searching for compounds that interact and successfully target or cross the membrane(s) of slow-growing or non-dividing bacteria [181]. Clustering the set of fluorophores accumulated by *E. coli* [128] and the antibiotics listed in this report (Figure 12) by chemical structure similarity points at a strategy to explore a potential antipersister pharmacopeia. The structural similarity of daptomycin to trypan blue, for instance, could enable the search for membrane-targeting antibiotics. Daptomycin is a cyclic lipopeptide and membrane-active antibacterial from *Streptomyces roseosporus*. However, due to the intrinsic tendency of membrane-active compounds to have cross toxicity with mammalian cells, the capacity to explore a wide chemical space by high-throughput chemical synthesis and screening is necessary to increase the odds of identifying compounds with the desirable specificity.

## 4. Materials and Methods 

The scientific literature database PubMed (https://pubmed.ncbi.nlm.nih.gov/) was searched with the following query: antibiotic AND persister AND bacteria AND English[lang] NOT review[ptyp]. The abstracts or titles (if no abstract was available) of a total of 9047 results up to 30 April 2020 were screened for the keywords (in any order): in vitro, persister, bacteria and (proportion or incidence or measure). This was carried out with a grep bash routine in Linux and iterative manual screening of selected papers. At this point, it became apparent that the great majority (over 90%) of publications (dating back to 1946 in PubMed) from the initial search were not relevant to this data compilation. They had vaguely used the terms ‘persistence’ or ‘persister’ for chronic infections and/or to denote certain pharmacokinetic properties of antibiotics and other medications under investigation. Publications on eukaryotic microorganisms (yeasts) (e.g., [182]) and Archaea (e.g., [183]) were also excluded.

A selection of 847 primary publications deemed relevant to the theme of antibiotic bacterial persisters were manually surveyed for quantitative data from work carried out in planktonic cultures. Numerical data were finally extracted from 187 articles [3,4,7,8,11,12,36,39,40,49,51,52,55,56,57,58,61,66,67,68,89,115,116,117,119,121,122,126,133,141,142,143,150,163,173,177,184,185,186,187,188,189,190,191,192,193,194,195,196,197,198,199,200,201,202,203,204,205,206,207,208,209,210,211,212,213,214,215,216,217,218,219,220,221,222,223,224,225,226,227,228,229,230,231,232,233,234,235,236,237,238,239,240,241,242,243,244,245,246,247,248,249,250,251,252,253,254,255,256,257,258,259,260,261,262,263,264,265,266,267,268,269,270,271,272,273,274,275,276,277,278,279,280,281,282,283,284,285,286,287,288,289,290,291,292,293,294,295,296,297,298,299,300,301,302,303,304,305,306,307,308,309,310,311,312,313,314,315,316,317,318,319,320,321,322,323,324,325,326,327,328,329,330,331,332,333,334] (Appendix A) published between 1970 and 2020 and reporting persister assays for antibiotic-sensitive bacteria strains. The data represented levels of persisters as percentages of cells growing as colony-forming units—cfu—on solid media after exposure to antibiotics for one or more time points in comparison with the sample at time zero. When the cfu post-antibiotic were not explicitly stated, numerical data were estimated by visual inspection of the relevant plots. The growth phase of the batch cultures from which the bacteria tested for persistence to antibiotics originated (i.e., exponential phase or stationary phase) was recorded as stated in each study. Numerical data were represented with three significant figures, with the exception of the very small quantities of the data for the lowest levels of persister.

Plotting and statistical analyses were carried out using R (https://www.r-project.org/). The hierarchical dendrogram and heatmap of the palette of 47 fluorophores were scripted with R’s *dendextend* and *ggplot*’s *heatmap.2* packages. The *hclust* function of *dendextend* produced hierarchical clusters from a Tanimoto similarity matrix derived as follows: cheminformatic (structural) fingerprints of 47 fluorophores were derived from their InChl notation using the Patterned algorithm within the RDkit (www.rdkit.org/) nodes in KNIME (http://knime.org/) [129,335]; a Tanimoto distance (TD) matrix was generated from these fingerprints (KNIME’s Distance Matrix Calculate); and the scores of this matrix were converted to Tanimoto similarity indices (1-TD) using KNIME’s Similarity Search.

The fitting of the time–kill data was carried out as implemented in R’s ggplot loess (local polynomial regression fitting) method. Local fitting uses the distance of data in the neighborhood of each dependent variable (time) to weight the least squares of the independent variable (count of cells of persisters after each time point of incubation). The size of the neighborhood is controlled by the ggplot’s span parameter in geom_smooth or stat_smooth. The default span applied here uses tricubic weighting (proportional to (1 − (distance/max distance)^3)^3) [336].

## 5. Conclusions

Bacteria can tolerate lethal concentrations of antibiotics with cells that are slow growers in an otherwise rapidly proliferating population. Deploying different mechanisms, bacteria undergoing growth-limiting conditions can also survive—in higher proportions (30-fold difference)—the presence of antibiotics. Species represented here are those clinically considered of high priority, usually those with a high incidence of antibiotic resistance (e.g., ESKAPE species), with *E. faecium* leading the table of antibiotic persisters. The most tested antibiotics (fluoroquinolones, penicillins, aminoglycosides, and cephalosporins) had similar persister levels of 0.12–0.16%. Lipopeptides (membrane-active compounds) were similarly represented but with half the levels of persisters, 0.05%. Two of the fluoroquinolones (ciprofloxacin and ofloxacin) and gentamicin exemplified here the Eagle effect, where higher concentrations of antibiotic can be less cytotoxic. This serves as a reminder of the need for optimization of bacteria–antibiotic concentration combinations before the surviving cells from time–kill assays can be interpreted as persisters. The cell wall of Gram-positive bacteria seems to give them an advantage point to tolerate bactericidal concentrations of antibiotics. Since most antibiotics are bactericidal against rapidly growing bacteria, not against slow-growing organisms, the activity seen by lipopeptides producing low levels of persister cells is encouraging. To this extent, we point at the strategy offered by fluorescence-based screening with a validated palette of fluorophores known to be accumulated in *E. coli*. The need for discovery platforms for antipersister antimicrobials cannot be overstated [22,23,337]. The road map from this sample of 50 years of quantitative bacterial persister data, rather clearly, is guiding us to strive for finding bactericidal compounds that can traverse or are active against bacteria-specific cell membranes.

## Figures and Tables

**Figure 1 antibiotics-09-00508-f001:**
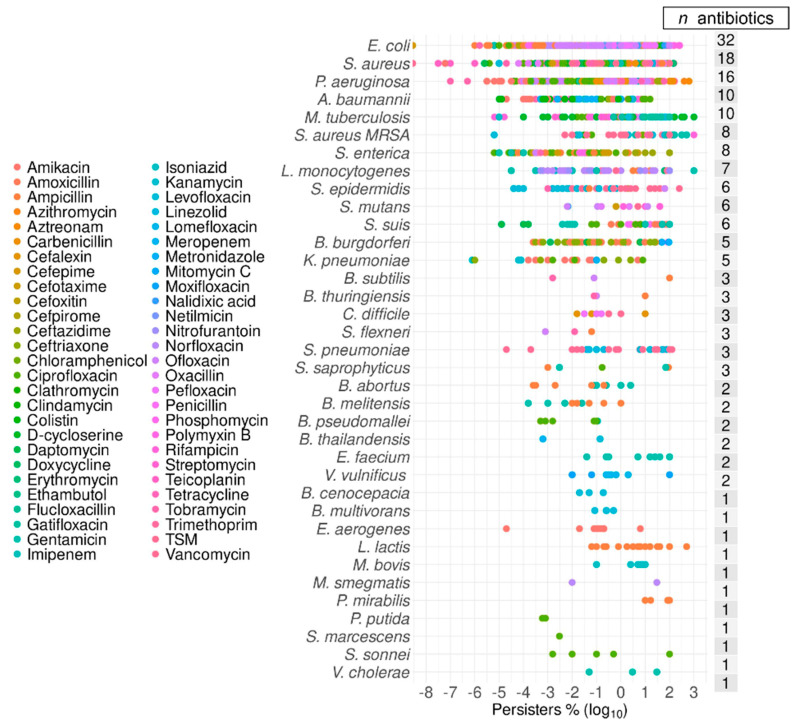
Antibiotics and bacterial species included in this survey. A total of 54 different antibiotics that were listed are represented by color-coded points along the list of 36 bacterial species. The bacteria were ordered according to the number of different antibiotics used in each species (*n* antibiotics). Most antibiotics were highly represented in the top four species. TSM is the antifolate mixture of trimethoprim and sulphamethoxazole.

**Figure 2 antibiotics-09-00508-f002:**
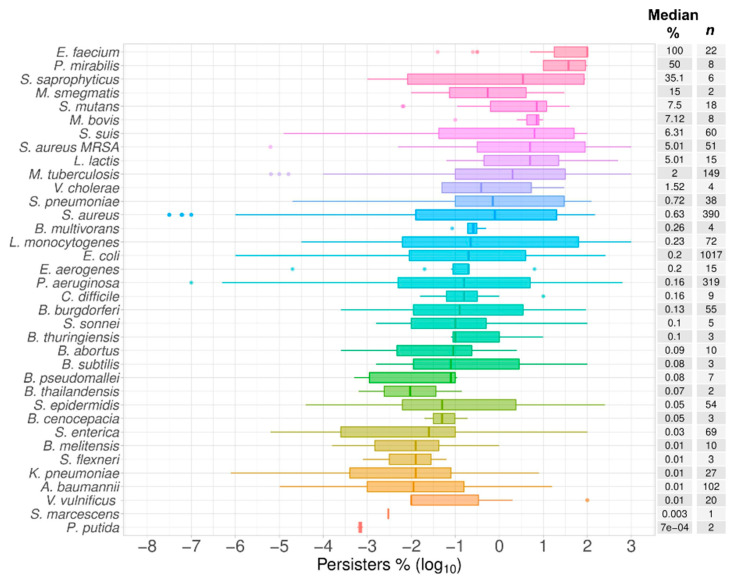
Bacterial species. A total of 36 bacterial species were listed in the order of the median value as plotted as box plots of the percentage of cells surviving the exposure to antibiotics in comparison to time zero in time–kill assays, or in comparison to the control without antibiotics in one-time-point experiments. The abscissa represents the spread of the data as log_10_ values of the percentage of persisters. The median value for the percentage of surviving cells is listed in the median % column followed the total number of time points collected for each species (*n*).

**Figure 3 antibiotics-09-00508-f003:**
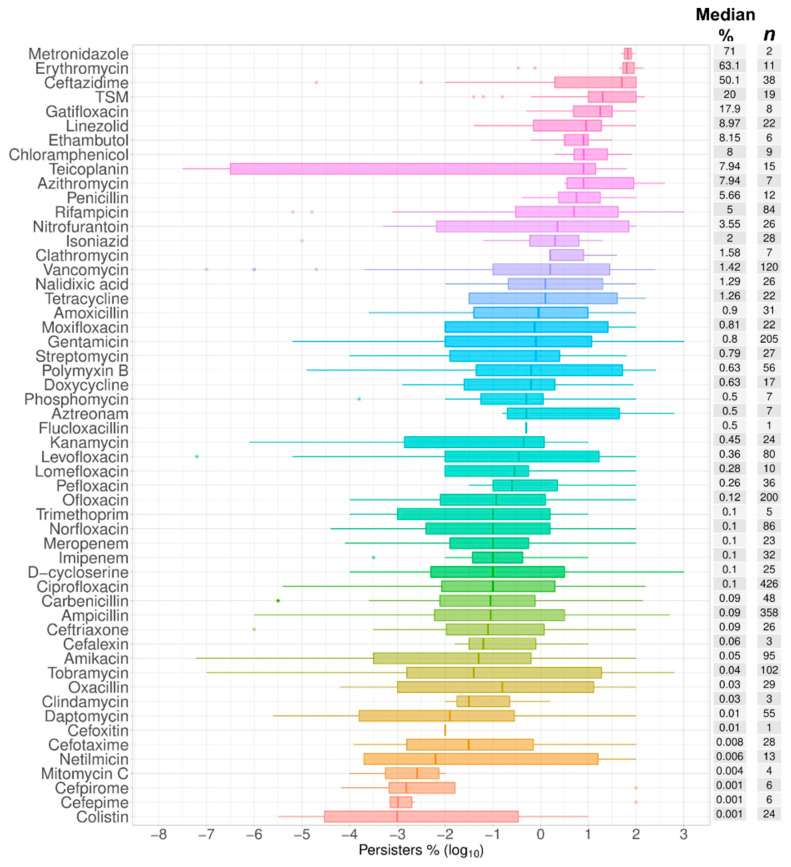
Antibiotics. A total of 54 antibiotics were listed in the order of the median value as plotted as box plots of the percentage of cells surviving the exposure to each of these antibiotics in comparison to time zero in time–kill assays, or in comparison to the control without antibiotics in one-time-point experiments. The abscissa represents the spread of the data as log_10_ values of the percentage of persisters. The median value for the percentage of surviving cells is listed in the median % column followed the total number of time points collected for each species (*n*).

**Figure 4 antibiotics-09-00508-f004:**
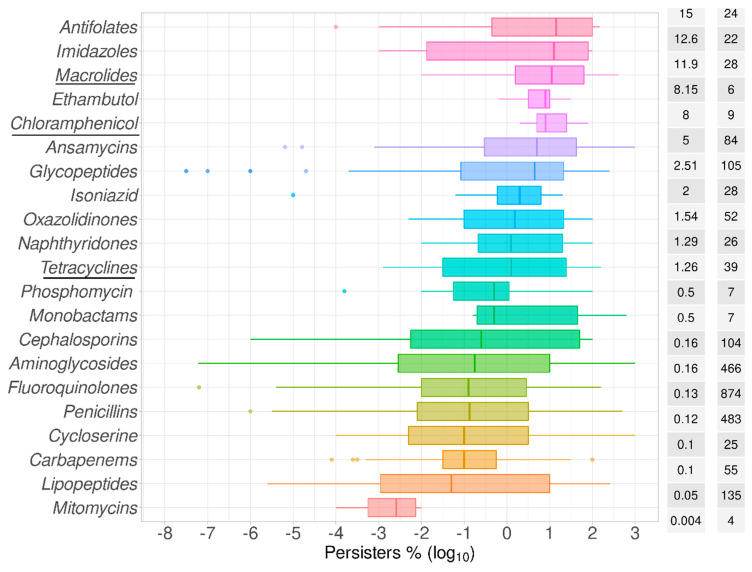
Persisters as compared by antibiotic classes. Antibiotics were grouped into 19 different classes according to their known chemical similarity (Table 2). These antibiotic classes were listed in the order of the median value as plotted as box plots of the percentage of cells surviving the exposure to each of these antibiotics in comparison to time zero in time–kill assays, or in comparison to the control without antibiotics in one-time-point experiments. Bacteriostatic antibiotics are underlined. The abscissa represents the spread of the data as log_10_ values of the percentage of persisters. The median value for the percentage of surviving cells is listed in the median % column followed by the total number of time points collected for each species (*n*).

**Figure 5 antibiotics-09-00508-f005:**
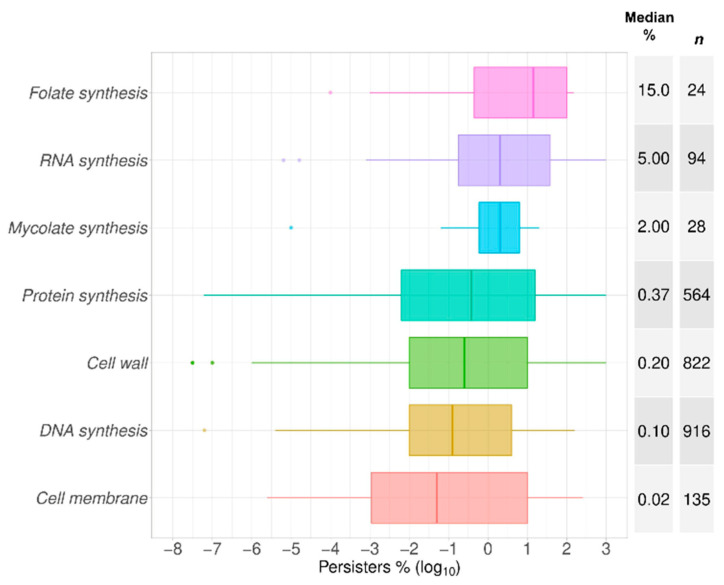
Bacterial persisters compared by antibiotic mechanism of action. Antibiotics were separated in 7 groups according to their main mechanism of action or target (Table 2). These groups were listed in the order of the median value as plotted as box plots of the percentage of cells surviving the exposure to each of these antibiotics in comparison to time zero in time–kill assays, or in comparison to the control without antibiotics in one-time-point experiments. The abscissa represents the spread of the data as log_10_ values of the percentage of persisters. The median value for the percentage of surviving cells is listed in the median % column followed by the total number of time points collected for each species (*n*).

**Figure 6 antibiotics-09-00508-f006:**
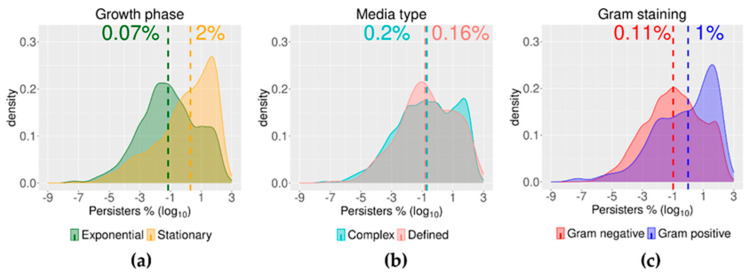
Data distribution for growth phase, media type, and Gram staining. (**a**) Growth phase: the data were clustered according to the growth phase of the cultures used for the persister assays. The abscissa represents the density distribution of the percentage of persister cells. The median of the persisters was 0.07% for cells from cultures in exponential growth phase and 2% for cells in stationary phase; (**b**) Media type: the data were clustered according to the media (either complex or defined media) of the cell cultures used for the challenge with antibiotics. The abscissa represents the density distribution of the percentage of persister cells. The median of the persisters was 0.2% for cells growing in complex media and 0.16% for cells growing in defined media; (**c**) Gram staining: the data as clustered according to the Gram-staining properties of all bacteria species. The abscissa represents the density distribution of the percentage of persister cells. The median of the persisters was 0.11% for cells from Gram-negative species and 1% for cells from Gram-positive species.

**Figure 7 antibiotics-09-00508-f007:**
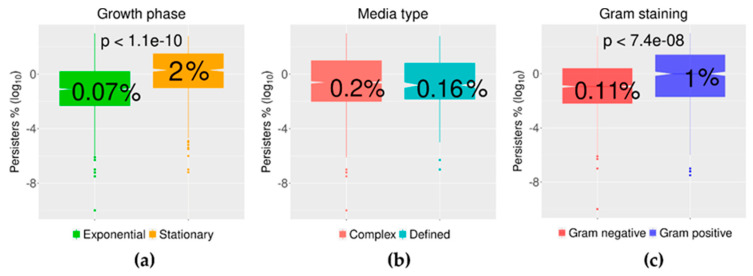
Significant differences for the data distribution for the growth phase and Gram staining but not media type. Growth phase: the box plots represent the distribution of the level of persisters as the log_10_ of the percentage of cells surviving the challenge of antibiotics. The median of the persisters was 0.07% for cells from cultures in the exponential growth phase and 2% for cells in the stationary phase. The significance of this difference was calculated by a two-tail Welch’s *t*-test (*p* < 1.1 × 10^−10^), which considers differences in the size of the samples compared. Media type: the data by media type (either complex or defined media) of the cell cultures used for the challenge with antibiotics showed no differences. Gram staining: data clustered by Gram staining. The difference in the median of the percentage of persisters was significant to *p* < 7.4 × 10^−8^.

**Figure 8 antibiotics-09-00508-f008:**
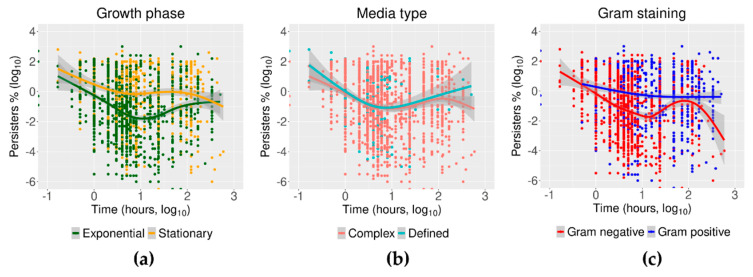
Time–kill curves for growth phase, media type, and Gram staining. The incubation time in the abscissa of each plot is represented as the log_10_ of time in hours. The incidence of persister cells is also the log_10_ of the percentage of cells growing in solid media after exposure to antibiotics in comparison to the number of cells at time zero. The number of persisters against time were fitted by local polynomial regression using weighted least squares as detailed in the methods. Growth phase: The growth-dependent data showed triphasic phases for persisters from both exponential and stationary cell cultures as described in the main text. Media type: The data grouped by media type showed no significant differences except for the third phase (after 100 h) observed in the cells from cultures in complex media. Gram staining: while the fitted data for persisters in Gram-positive bacteria had a continuously descending trend, Gram-negative bacteria had a trimodal curve as described.

**Figure 9 antibiotics-09-00508-f009:**
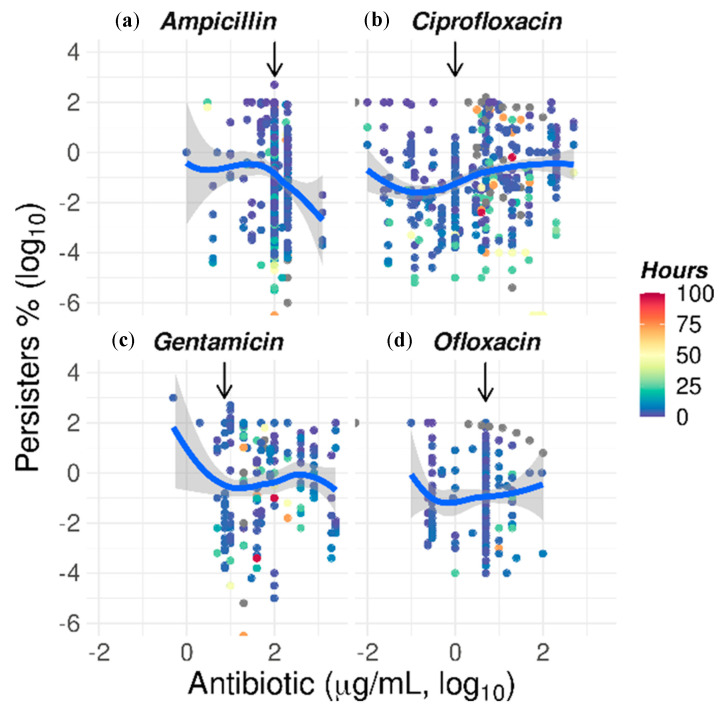
Effect of antibiotic concentrations on the levels of bacteria persistent to antibiotics: The concentration of antibiotic is represented as the log_10_ of µg/mL along the abscissa of each plot. The incidence of persister cells is represented also as the log_10_ of the percentage of cells growing in solid media after exposure to antibiotics in comparison to the number of cells at time zero. The number of persisters against time were fitted by local polynomial regression fitting by weighted least squares as detailed in the materials and methods. The incubation times for any given concentrations of antibiotic were also plotted with a color spectrum of blue to red for up to 100 h. The arrows point at the most frequently used concentration for each antibiotic (mode of antibiotic usage in Appendix A).

**Figure 10 antibiotics-09-00508-f010:**
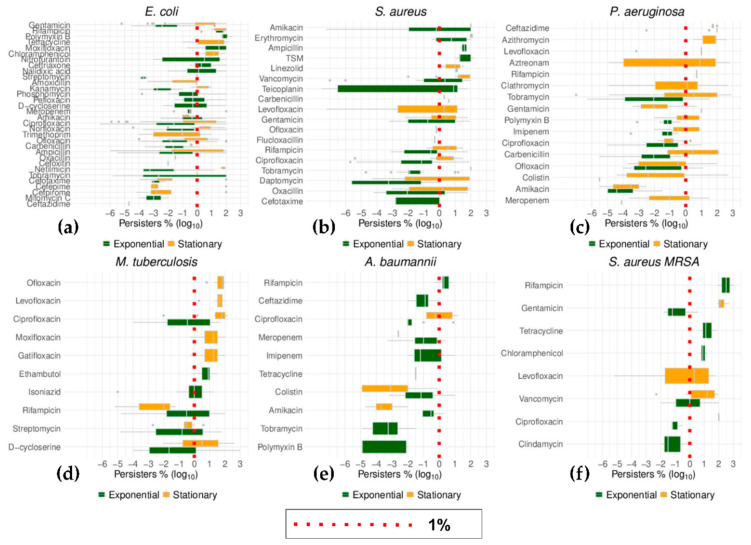
Distribution of the data from the six most studied bacteria species as grouped by growth phase. The percentages (as log_10_) of persisters to antibiotics as listed for each species were ordered by their median values. The growth phase of the cultures that underwent persistence tests were differentially plotted, green for samples from exponential cultures and orange for stationary cultures. The red dotted lines represent the position at which levels of persister cells of 1% are in each box plot. With the exception of the data for colistin in *A. baumannii*, the higher number of persisters found in cells from stationary cultures exemplified here in the six most commonly studied bacteria species.

**Figure 11 antibiotics-09-00508-f011:**
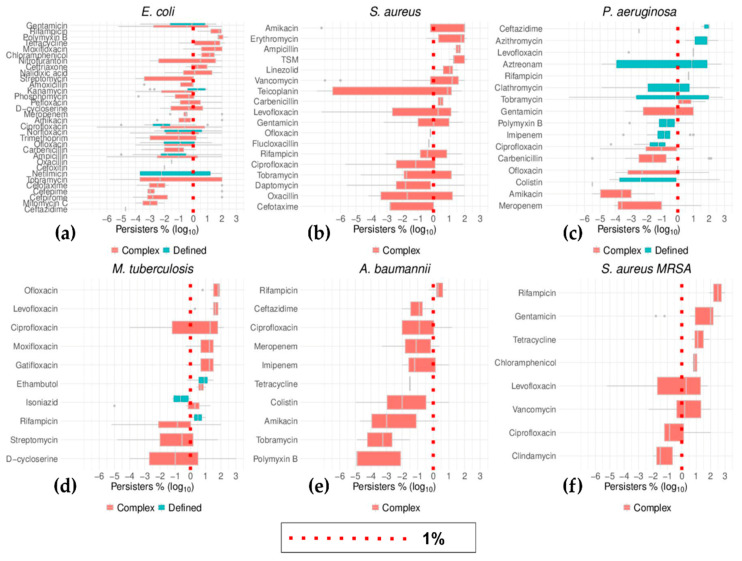
Distribution of the data grouped by media type in the six most studied bacteria species. The percentage (as log_10_) of persisters to antibiotics as listed for each species were ordered by their median values. The media type of the cultures that underwent persistence tests were differentially plotted, red for samples from cultures in complex media and green for samples from cultures in defined media. The red dotted lines represent the position at which levels of persister cells of 1% are in each box plot. As elaborated in the main text, there were differences in the incidence of persisters between cells from either type of growth media. However, it is obvious that the data from either media are not equally represented, with defined media sparely used in only *E. coli*, *P. aeruginosa*, and *M. tuberculosis*.

**Figure 12 antibiotics-09-00508-f012:**
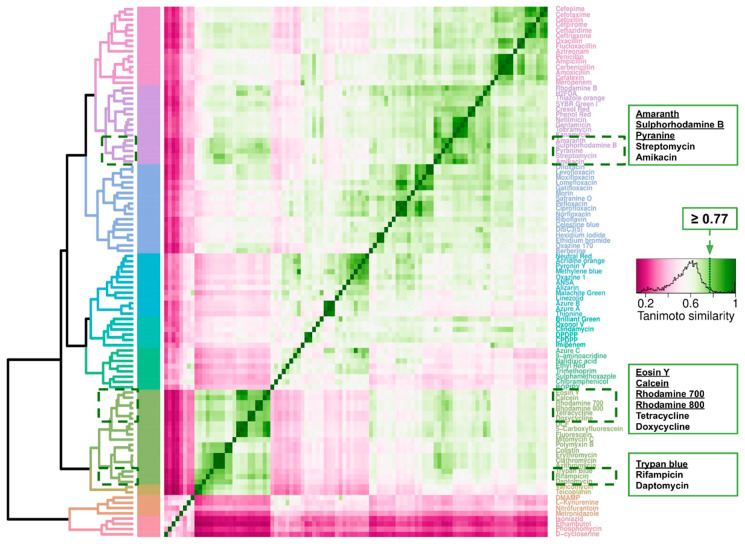
Chemical structure similarity of the antibiotics compiled in this study and some fluorescent molecules known to accumulate in bacteria. Heatmap and dendrogram representing the Tanimoto similarities (derived as described in the methods) of 47 fluorescence compounds known to accumulate in *E. coli* [129] and the 54 antibiotics listed in this paper. Scale from zero (least similarity) to 1 (highest similarity). Dotted boxes indicate clusters with a Tanimoto similarity greater than or equal to 0.77 (a cut-off for similar bioactivities is accepted at about 0.8). The three clusters where antibiotics and fluorophores have this level of structural similarity are re-written in framed boxes with the non-antibiotic fluorophores underlined in each case.

**Table 1 antibiotics-09-00508-t001:** Bacterial species. List of the 36 bacteria species that appeared in the publications included in this survey. For *Enterobacter aerogenes*, the name was kept as reported in the original publications although it is currently named *Klebsiella aerogenes*. *S. enterica* was serotype Typhimurium. The list was separated according to the Gram-staining properties of these bacteria with 20 species listed as Gram-negative and 16 as Gram-positive. (*) *Bacillus* spp. and *Clostridium* spp. are known to form spores. The persistence to antibiotics data for these spore-forming bacteria originated from work carried out in planktonic cultures where these bacteria were germinating.

Gram Staining	
Negative	Positive
*Acinetobacter baumannii*	*Bacillus subtillis* (*)
*Borrelia burgdorferi*	*Bacillus thuringiensis* (*)
*Brucella abortus*	*Clostridium difficile* (*)
*Brucella melitensis*	*Enterococcus faecium*
*Burkholderia cenocepacia*	*Lactobacillus lactis*
*Burkholderia multivorans*	*Listeria monocytogenes*
*Burkholderia pseudomallei*	*Mycobacterium bovis*
*Burkholderia thailandensis*	*Mycobacterium smegmatis*
*Enterobacter aerogenes*	*Mycobacterium tuberculosis*
*Escherichia coli*	*Staphylococcus aureus*
*Klebsiella pneumoniae*	*Staphylococcus aureus MRSA*
*Proteus mirabilis*	*Staphylococcus epidermidis*
*Pseudomonas aeruginosa*	*Staphylococcus saprophyticus*
*Pseudomonas putida*	*Streptococcus mutants*
*Salmonella enterica*	*Streptococcus pneumoniae*
*Serratia marcescens*	*Streptococcus suis*
*Shigella flexneri*	
*Shigella sonnei*	
*Vibrio cholera*	
*Vibrio vulnificus*	

**Table 2 antibiotics-09-00508-t002:** Antibiotic classes. The first column lists the primary mechanism of action or target. The class groups some of the antibiotics based on their known chemical similarity. We do not include mixtures containing, e.g., the penicillinase inhibitor clavulanic acid. The count of antibiotics reached 54, which included one mixture of antifolates in trimethoprim-sulfamethoxazole (TSM). In some publications, this mixture was denoted as Co-trimazol, which here was included in the TSM tally. (*) Antibiotics considered bacteriostatic that were included here based on their reported bactericidal activity: chloramphenicol [74,75,76], azithromycin [77,78,79], clathromycin [78,80,81,82], clindamycin, erythromycin [82,83,84,85], doxycycline [86,87], and tetracycline [86,88]. The rest of the antibiotics are considered primarily bactericidal.

Mechanism of Action	Class	Antibiotic
Cell Wall Synthesis	Carbapenems	Imipenem
		Meropenem
	Cephalosporins	Cefalexin
		Cefepime
		Cefotaxime
		Cefoxitin
		Cefpirome
		Ceftazidime
		Ceftriaxone
	Cycloserine	D-cycloserine
	Ethambutol	Ethambutol
	Glycopeptides	Teicoplanin
		Vancomycin
	Monobactams	Aztreonam
	Penicillins	Amoxicillin
		Ampicillin
		Carbenicillin
		Flucloxacillin
		Oxacillin
		Penicillin
	Phosphomycins	Phosphomycin
Protein synthesis	Aminoglycosides	Amikacin
		Gentamicin
		Kanamycin
		Netilmicin
		Streptomycin
		Tobramycin
	Chloramphenicol (*)	Chloramphenicol
	Macrolides (*)	Azithromycin
		Clathromycin
		Clindamycin
		Erythromycin
	Oxazolidinones	Linezolid
	Tetracyclines (*)	Doxycycline
		Tetracycline
DNA synthesis	Fluoroquinolones	Ciprofloxacin
		Gatifloxacin
		Levofloxacin
		Lomefloxacin
		Moxifloxacin
		Norfloxacin
		Ofloxacin
		Pefloxacin
	Imidizoles	Metronidazole
		Nitrofurantoin
	Mitomycins	Mitomycin C
	Naphthyridones	Nalidixic acid
Cell membrane	Lipopeptides	Colistin
		Daptomycin
		Polymyxin B
Folate synthesis	Antifolates	Trimethoprim
		TSM
Mycolic acid synthesis	Isoniazid	Isoniazid
RNA synthesis	Ansamycins	Rifampicin

**Table 3 antibiotics-09-00508-t003:** Descriptive statistics for three of the variables measured for the incident of persisters in bacteria: The median, mean, and confidence interval 95% (CI95) were listed for the data grouped by growth phase, Gram staining, and growth media. The total number of data points collected from source are denoted as *n*.

	Growth Phase		Growth Media Type		Gram Staining	
	Exponential	Stationary	Complex	Defined	Negative	Positive
**Median (%)**	0.07	2.00	0.20	0.16	0.11	1.00
**Mean (%)**	12.6	26.8	17.2	19.9	12.7	27.1
**CI95 (%)**	10.0–15.1	23.3–30.3	15.1–19.4	12.8–27.0	10.9–14.5	22.2–32.0
***n***	1675	908	2248	335	1707	876

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
