# Peer review of "A Quantitative Survey of Bacterial Persistence in the Presence of Antibiotics: Towards Antipersister Antimicrobial Discovery"

_antibiotics, 2020, doi:10.3390/antibiotics9080508_

Round 1

Reviewer 1 Report

Dear Authors,

I believe that the work is valuable, brings a new point of view, and interesting information to the field of antibiotic application. You deserve recognition for the enormous amount of work they put into the analysis of such a large number of publications devoted to the influence of various antibiotics to bacterial survival.

Below are only minor suggestions:

  • page 3, line 111 – the comment refers to the incubation time - neither here, nor in the supplementary material S1, it was stated in which units the time was measured. Was it in minutes or hours ... - please specify it in the text or in the description of table S1.
  • page 11, Table 3 - table content is doubled - is it by mistake?
  • The caption for Figure 10 is in the text on page 16, lines 400-406, and should be below the graph.
  • Figures 10 and 11, pages 15 and 16 - In both cases, the names of the antibiotics under a) are very vague. Maybe the chart should be placed across the page?

Author Response

Reviever 1:

Dear Authors,

I believe that the work is valuable, brings a new point of view, and interesting information to the field of antibiotic application. You deserve recognition for the enormous amount of work they put into the analysis of such a large number of publications devoted to the influence of various antibiotics to bacterial survival.

Below are only minor suggestions:

page 3, line 111 – the comment refers to the incubation time - neither here, nor in the supplementary material S1, it was stated in which units the time was measured. Was it in minutes or hours ... - please specify it in the text or in the description of table S1.

Authors: the time unit (hours) has been specified in the main text and in the legend to Table S1.

page 11, Table 3 - table content is doubled - is it by mistake?

Authors: Yes, unintended error. Thank you.

The caption for Figure 10 is in the text on page 16, lines 400-406, and should be below the graph.

Authors: Relevant layout corrected. Thank you.

Figures 10 and 11, pages 15 and 16 - In both cases, the names of the antibiotics under a) are very vague. Maybe the chart should be placed across the page?

Authors: the density of the graphic format (png) for these figures has been increased and re-inserted in the main text. The journal will also have the pdf (with much higher quality) versions for all the figures of this paper for the editorial process that should deliver a post-review manuscript with publication-quality figures.

Reviewer 2 Report

The threat of antibiotic-resistant bacteria is increasing worldwide. Bacteria utilize persistence and resistance to survive antibiotic stress. Bacterial persisters are rare phenotypic variants that are temporarily tolerant to high concentrations of antibiotics. In the current review article Salcedo-Sora et al. have compiled and analyzed persistence related data from 187 manuscripts and evaluated the impact of several factors, such as the class of antibiotics and bacterial type. Authors have done a great job in compiling the knowledge.  I find this manuscript suitable for acceptance after making some minor changes.

  1. Several figures can be combined such as Figure 2 and Figure 3, and so on.
  2. It would be great to include a figure demonstrating all suspected molecular pathways that could cause the emergence of persistence.
  3. Please check typos.

Author Response

Reviewer 2:

The threat of antibiotic-resistant bacteria is increasing worldwide. Bacteria utilize persistence and resistance to survive antibiotic stress. Bacterial persisters are rare phenotypic variants that are temporarily tolerant to high concentrations of antibiotics. In the current review article Salcedo-Sora et al. have compiled and analyzed persistence related data from 187 manuscripts and evaluated the impact of several factors, such as the class of antibiotics and bacterial type. Authors have done a great job in compiling the knowledge. I find this manuscript suitable for acceptance after making some minor changes.

Several figures can be combined such as Figure 2 and Figure 3, and so on.

Authors: all the figures in this paper are high-content graphics, and already densely populated. The suggested combination of figures would severely impact their readability and understanding.

It would be great to include a figure demonstrating all suspected molecular pathways that could cause the emergence of persistence.

Authors: this is a great visualisation challenge which has been undertaken by approximately 17 reviews between 2007 and 2020. We trust that the 13 reviews referenced in our paper can convey such complex scenarios. We are confident that the results of our own manuscript have been visually and clearly conveyed in the 12 figures presented.

Please check typos.

Authors: the entire manuscript has been checked for typos. Thank you.